# The Comparison of Honeybee Viral Loads for Six Honeybee Viruses (ABPV, BQCV, CBPV, DWV, LSV3 and SBV) in Healthy and Clinically Affected Honeybees with TaqMan Quantitative Real-Time RT-PCR Assays

**DOI:** 10.3390/v13071340

**Published:** 2021-07-11

**Authors:** Laura Šimenc, Tanja Knific, Ivan Toplak

**Affiliations:** 1Virology Unit, Institute of Microbiology and Parasitology, Veterinary Faculty, University of Ljubljana, Gerbičeva 60, 1115 Ljubljana, Slovenia; ivan.toplak@vf.uni-lj.si; 2Institute of Food Safety, Feed and Environment, Veterinary Faculty, University of Ljubljana, Gerbičeva 60, 1115 Ljubljana, Slovenia; tanja.knific@vf.uni-lj.si

**Keywords:** honeybees, viral load, monthly sampling, RT-qPCR, prevalence

## Abstract

The viral loads of acute bee paralysis virus (ABPV), black queen cell virus (BQCV), chronic bee paralysis virus (CBPV), deformed wing virus (DWV), Lake Sinai virus 3 (LSV3), and sacbrood bee virus (SBV) were determined in samples with the use of quantitative TaqMan real-time reverse transcription and polymerase chain reaction (RT-qPCR). A total of 108 samples of healthy adult honeybees from four differently located apiaries and samples of honeybees showing different clinical signs of viral infections from 89 apiaries were collected throughout Slovenia. The aim of this study was to discover correlations between viral loads and clinical signs in adult honeybees and confirm previously set threshold viral load levels between healthy and clinically affected honeybees. Within this study, two new RT-qPCR assays for quantification of LSV3 and SBV were developed. Statistically significant differences in viral loads of positive samples were identified between healthy and clinically affected honeybees for ABPV, CBPV, DWV, and SBV, while for BQCV and LSV3, no statistical differences were observed between both groups. Despite high detected LSV3 prevalence and viral loads around 6.00 log_10_ viral copies/bee, this lineage probably has a limited impact on the health status of honeybee colonies. The determined viral loads between 3.94 log_10_ and 13.17 log_10_ in positive samples for six viruses, collected over 10 consecutive months, including winter, present additional information of high viral load variations in healthy honeybee colonies.

## 1. Introduction

Viruses are important pathogens of honeybees (*Apis mellifera*) frequently detected in commercial and hobby beekeeping apiaries, while different viruses can also infect other wild pollinators, such as bumblebees, for which the pathogenicity of viral infections is still not fully understood [1,2,3,4]. More than 30 honeybee viruses have been identified and described using different molecular methods, including next-generation sequencing (NGS) [5,6]. Most honeybee viral infections might be present in subclinical form, although in the combination of several factors, such as insufficient feeding of honeybee colonies, high varroa mite (*Varroa destructor*) infestations, or the presence of bacterial infections, the honeybee viruses can significantly contribute to honeybee losses [7].

Most pathogenic honeybee viruses are about 30 nm small isometric particles containing a single-strand positive RNA molecule [8]. Sacbrood bee virus (SBV) and deformed wing virus (DWV) are assigned to genus Iflavirus (family *Iflaviridae*), acute bee paralysis virus (ABPV) and black queen cell virus (BQCV) are classified as Cripavirus (family *Dicistroviridae*), while chronic bee paralysis virus (CBPV) remains unclassified [9]. Lake Sinai virus (LSV) is an RNA virus, taxonomically classified as Sinaivirus genus, genetically related to chronic bee paralysis virus (CBPV) and viruses from the *Nodaviridae* family [10]. These viruses are the most frequently detected viruses in honeybee colonies, and they can infect larvae, pupae, and adult bees; the exception is CBPV, which only infects adult bees [11,12].

The infection with ABPV in honeybee colonies might not be apparent, while the fatal infections are usually related to *V. destructor* infestation, causing paralysis of adult bees and dead brood in a short period [13]. Although the number of sequenced strains worldwide is increasing and genetically diverse ABPV strains have been identified, the virulence of different strains remains poorly understood. The nucleotide comparison of 54 ABPV-positive samples from Slovenia revealed two distinct genetic clusters (clusters 1 and 2) in ORF 1 and ORF 2 gene regions, and ABPV strains from cluster 2 were associated with more severe clinical symptoms [14].

BQCV is one of the most prevalent and widespread viruses in honeybee colonies, with a constant annual incidence in the adult population closely related to *Nosema apis* or *Nosema ceranae* infestation [15]. Although BQCV does not cause visible symptoms in infected adult bees, it can kill developing queen larvae turning their cells black [8,15]. According to reports of beekeepers, BQCV can cause problems with queen rearing and may influence over-wintering honeybee losses [16].

CBPV shows severe infections, which appear sporadically, causing severe paralyses in adult bees and may contribute to colony losses. The viral outbreaks are often reported in the spring and summer months. There are indications that over-wintering losses may be associated with CBPV infection [16,17]. The clinical manifestation of disease is severe, inducing trembling and cuticle melanisation (black bees) before causing the death of numerous foragers in front of the hive entrance; some clinical signs such as trembling may be confused with intoxication syndrome [18]. CBPV and *N. ceranae* experimental co-infection of winter honeybees revealed replication of virus after inoculation of experimentally caged bees with field CBPV inoculum. The experimental infection confirmed that either CBPV alone or combined with *N. ceranae* is an important pathogen of honeybees [19].

The incidence of disease caused by DWV is strongly co-related with colonies infested with *V. destructor*, although the losses of adult workers; queens; brood; and, finally, whole honeybee colonies have been implicated in the absence of the varroa mite. DWV causes wing deformities during the developing stages of honeybees. The infected honeybees may die shortly, and DWV thus reduces the lifespan of infected bees [20,21,22]. Phylogenetic analyses of DWV strains show that several of the identified DWV strains from honeybees and varroa mites are genetically identical or very closely related, confirming the significant role of *V. destructor* in the DWV transmission [23].

LSV was first described as the most abundant pathogen detected in a monitoring study conducting in the USA in 2008–2009 [10]. The pathogenicity of LSV is not yet well understood, but this virus could be one of the co-factors contributing to collapsing honeybee colonies [10,24,25,26,27]. LSV replicates in honeybees that often harbour high viral burdens (10^8^–10^9^ viral copies/bee) [5]. Although there are no obvious clinical symptoms attributed to LSV-positive honeybee families, their abundance and association with colonies of poor health are indications that different genetic strains of LSV might have a negative impact on honeybee health [28].

SBV is a honeybee virus that may persist in adult honeybee and other wild pollinator populations as an asymptomatic infection decreasing their lifespan [29]. Clinically most affected are larvae, while after their infection with SBV, they stop pupating, and fluid accumulates between the larva’s body and unshed skin, forming a saccule before they die [29].

The key diagnostic parameter of viral infections in honeybee colonies is not only the presence of a certain virus but also its viral load [30]. The viral loads may be variable throughout the season but also between different honeybee categories [31]. The importance of determining the viral loads and possibly setting the threshold limits of causing clinical effects for each virus are extremely important to monitor honeybee health status and set correct diagnosis in healthy, affected, or treated honeybee colonies. Compared to the conventional RT-PCR methods, the quantitative RT-qPCR assays with primers, TaqMan probes, and quantification standards provide more efficient, sensitive, specific, and accurate diagnostics of viral infections in honeybee colonies. Nevertheless, these methods are still not widely implemented in honeybee diagnostic laboratories [18,30,32,33].

The first comprehensive survey from 2002 made in France reveals the prevalence and seasonal variations of six different viruses detected with the use of conventional RT-PCR assays in adult honeybees and the pupae of seemingly healthy colonies. In adult bees, ABPV was found in 58% of the apiaries with a higher incidence in the summer and autumn, while BQCV was found in 86% of the apiaries. CBPV was detected only in adult bees in 28% of the apiaries with no exact seasonal pattern; DWV was by far the most frequently detected virus (97%), but SBV was also found in the majority of the apiaries (86%). The study indicates that bee viruses can occur persistently despite the lack of clinical signs, suggesting that disease outbreaks might be activated by environmental factors [13].

The seasonal dynamics of four honeybee viruses in Slovenia in pupae, hive, and forager bee samples tested with conventional RT-PCR assays provide evidence of horizontal transmission of ABPV, BQCV, and DWV, which occur through contact between social groups from forager to hive bees and pupas, where the lowest prevalence for each of four tested honeybee viruses was detected in pupas [31].

This study aimed to compare the prevalence and viral loads for six different honeybee viruses (ABPV, BQCV, CBPV, DWV, LSV3, and SBV) in tested samples of adult honeybees with no clinical signs, collected monthly in four different locations over a one-year period and in clinically affected adult honeybees. Two newly developed RT-qPCR assays for the detection of LSV3 and SBV were evaluated for the first time and tested on field samples of healthy and clinically affected adult honeybees.

## 2. Materials and Methods

### 2.1. Sample Collection

One hundred and eight honeybee samples were collected from healthy honeybee colonies during 10 months (from October 2018 to October 2019) of consecutive sampling. In four different apiaries, three honeybee colonies were selected in each apiary, located in different geographical areas of Slovenia (Location 1—Štajerska, Location 2—Prekmurje, Location 3—Gorenjska, Location 4—Ljubljana) (Figure 1). The apiary in Location 1 was owned by a large, private, intensively managed beekeeping organisation with 20 apiaries in different locations and more than 2000 honeybee colonies. This specific apiary for sampling consists of 100 honeybee colonies. In Location 2, a small hobby apiary was selected, consisting of 7 honeybee colonies, but well managed and with constant veterinary supervision. The apiary in Location 3 was an extensive solitary apiary with 50 honeybee colonies surrounded by well-preserved nature, about 15 km from other apiaries. The apiary in Location 4 was a school apiary owned by the Veterinary Faculty, University of Ljubljana, and is in the city centre of Ljubljana, with a high frequency of educational and scientific experiments. During each sampling, 10 live adult Carniolan honeybees (*Apis mellifera carnica*) were collected with sterile tweezers from individual honeybee colonies. Bees were stored in sterile plastic bags in a portable refrigerator and then frozen at −70 °C until further investigation. In specific months, sampling was not possible due to bad weather conditions that disturbed the honeybee colonies (Appendix A).

Between April 2016 and September 2020, 89 apiaries were selected throughout Slovenia by veterinary specialists. Clinically affected honeybee colonies with suspicion of viral infections, such as dead honeybees in front of the hives, colonies collapsing, weak colonies, trembling, paralysis, loss of colour and hair, wing deformities, or shorter life span, were observed. The 89 samples, each consisting of at least 10 affected honeybees *(Apis mellifera carnica*) from the same apiary, were collected and stored at –70 °C until further investigation (Figure 1).

### 2.2. Sample Preparation

Each sample consisting of 10 honeybees was placed into Ultra-Turrax DT-20 tubes (IKA, Königswinter, Germany) with 5 mL of RPMI 1640 medium (Gibco, Paisley, UK) and incubated at room temperature for 30 min. Samples were homogenised and centrifuged for 15 min at 2500× *g*; 2 mL of supernatant was stored from each sample as a suspension for further viral RNA extraction.

### 2.3. Primers, TaqMan Probes, and Standards Design

Primers, TaqMan probes, and quantification standards for ABPV [32], BQCV [34], CBPV [12], and DWV [30] were used from previously published protocols (Table 1 and Table 2). The new RT-qPCR protocols for detection and quantification of LSV3 and SBV were developed within the purpose of this study. Primers, TaqMan probes, and quantification standards were designed from multiple alignments of previously sequenced LSV3 and SBV strains from Slovenia [4,28] and around the world, available in GenBank. In the highly conserved polyprotein gene region for SBV and RNA-dependent RNA polymerase gene region for LSV3, primers, probes, and standards were designed using Geneious Prime software suite v. 11.0.5 (Biomatters Ltd., Auckland, New Zealand). The primers, probes and standards were produced by the company Sigma-Aldrich (Steinheim, Germany) (Table 1 and Table 2).

The intercept, slope, efficiency, and R^2^ of six different quantitative real-time RT-PCR assays were assessed via three repetitions of each method using 10-fold serial dilutions of the standard; from the obtained results, the mean value and standard deviation (SD) values were calculated (Table 3).

### 2.4. RNA Extraction and RT-qPCR Assays

The total RNA was extracted from 140 µL of suspension from each sample using the QIAamp viral RNA mini-kit (Qiagen, Hilden, Germany) and recovered from a spin column in 60 µL of elution buffer. Reverse transcription with RT-qPCR assay was made in a single step using QuantiNova Pathogen +IC Kit (Qiagen, Hilden, Germany). The RT-qPCR mix consisted of 5 µL QuantiNova Master Mix, 2 µL 10×Internal Control (IC) Probe Assay, 1 µL IC (1:100), 4.5 µL deionised water, 1 µL forward primer (200 nM), 1 µL reverse primer (200 nM) and 0.5 µL probe (100 nM), and 5 µL of extracted RNA with a total of 20 µL of the final volume. Thermal cycling was performed on an Mx3005P thermocycler (Stratagene, La Jolla, CA, USA) with the following conditions: 20 min at 50 °C, 2 min at 95 °C, followed by 45 cycles of 15 s at 95 °C, 30 s at 60 °C, and 30 s at 60 °C. Into each run, the positive control was included, prepared as a mixed suspension of previously determined positive field samples of six different viruses (ABPV, BQCV, CBPV, DWV, LSV3, and SBV). A positive control sample was further divided into 50 aliquots of 160 µL, and a positive result was expected for an individual virus with a quantitation cycle value (Cq) around 20. A negative control was prepared and used in the same way as a positive; each negative control consisted of only 160 µL of RPMI 1640 medium (Gibco, Paisley, UK) in aliquot.

The standard for each virus was constructed artificially using plasmid vector with a known viral RNA copy number. Tenfold dilutions of the standards from 10^−3^ to 10^−7^ were prepared and added to each RT-qPCR run. The exact number of RNA viral molecules in individual samples were determined from the standard curve for each of six honeybee viruses individually.

Results for each sample were analysed using MxPro-Mx3005P v4.10 software (Stratagene, La Jolla, CA, USA), and the exact copy number was determined. Firstly, results were expressed as the number of detected viral copies in 5 µL of extracted RNA. Each result was then calculated to log_10_ average copy number per bee using the following equation: average copy number per bee = copies/5 µL extracted RNA × 0.5 (10 honeybees are homogenised in 5 mL of RPMI) × 14.29 (140 µL from 2 mL of honeybee supernatant for RNA extraction) × 12 (5 µL from 60 µL of total extracted RNA for in qPCR mix) = 85.74; log_10_ (85.74) = 1.93. The determined conversion factor (1.93) was calculated according to previously published recommendations [18] and to the volumes used in RT-qPCR assays in this study. Each value for which log_10_ was lower than 2, which means approximately 100 viral copies/bee, was interpreted as a negative result and not included in further calculations of positive samples.

### 2.5. Statistical Analysis

Data handling was carried out in Microsoft Excel 365. Statistical analysis and figures were performed using R statistical software, version 3.6.2 [38]. Associations between the prevalence of ABPV, BQCV, CBPV, DWV, LSV3, and SBV in healthy and clinically affected honeybee colonies was compared using Fisher’s exact test [39]. The difference in viral loads of positive samples between healthy and clinically affected honeybee colonies was tested with the Wilcoxon rank-sum test [40], since the data were not normally distributed. *p*-values <0.05 were considered to be statistically significant, and *p*-values ≥0.05 and <0.1 as marginally significant.

## 3. Results

### 3.1. The Performance of RT-qPCR Assays for Six Honeybee Viruses

The performance of six different RT-qPCR assays designed and used for quantification of six honeybee viruses (ABPV, BQCV, CBPV, DWV, LSV3, and SBV) in healthy and clinically affected honeybee colonies is repetitive and accurate for the presentation of the results. The mean efficiency and R^2^ stand as follows: for ABPV: 105.9 ± 6.9 and R^2^: 0.985 ± 0.010; for BQCV: 107.6 ± 8.4 and R^2^: 0.996 ± 0.004; for CBPV: 94.9 ± 4.6 and R^2^: 0.997 ± 0.003; for DWV: 108.9 ± 9.37 and R^2^: 0.992 ± 0.011; for LSV3: 93.8 ± 3.9 and R^2^: 0.997 ± 0.002; for SBV: 94.7 ± 7.8 and R^2^: 0.999 ± 0.002 (Table 3). Two newly developed and implemented quantitative RT-qPCR assays for detecting and quantifying LSV3 and SBV showed 100% reliable results on 50 previously tested samples for each virus, which were confirmed positive by Sanger sequencing [4,14,28]. These two assays are set to be on the same level as previously described methods of the European Union Reference Laboratory for Honeybee Health (EURL ANSES, Sophia Antipolis, France) [38]. The determined mean R^2^ s of six different implemented real-time RT-PCR assays were from 0.985 to 0.999 (Table 3).

### 3.2. Viral Prevalence and Comparison of the Viral Loads from Samples of Healthy and Clinically Affected Honeybee Colonies

High variations in the 10-month prevalence were observed when comparing results between collected samples of healthy honeybee colonies in Locations 1, 2, 3, and 4 (Table 4). The lowest ABPV prevalence was detected in Location 1 (3.70%) and the highest in Location 4 (66.67%). The prevalence of BQCV varied from 96.30 to 100% in four locations (Table 4). No CBPV positive sample was found in healthy honeybee colonies. The lowest DWV prevalence was detected in Location 2 (4.76%) and the highest in Location 4 (70.00%). The LSV3 virus was detected in all four locations, with 46.67% prevalence in Location 3, while in Locations 1, 2, and 4, the prevalence was from 71.43 to 80.00%. The SBV virus prevalence varied between four locations (from 3.70 to 66.67%). The viral prevalence was strongly dependent on sampling location, with the highest prevalence for most viruses in Location 4 and lowest in Locations 1 and 3 (Table 4). No inhibitors of the kit’s inhibition control in any tested samples by RT-qPCR assays were detected.

The prevalence of ABPV-positive samples in affected honeybees (79.78%) is statistically significantly higher (*p* <0.0001) than in healthy honeybees (29.63%) (Figure 2). The ABPV viral load with a mean value of 5.98 log_10_ copies/bee for healthy was lower than the detected mean value of 7.21 log_10_ viral copies/bee for affected honeybees (*p* = 0.0284). High BQCV prevalence in healthy (98.15%) and affected honeybees (96.63%) was detected (*p* = 0.6597). No statistically significant difference between both groups was observed for BQCV with a mean 6.88 log_10_ viral copies/bee in healthy and 7.33 log_10_ viral copies/bee in affected honeybees (*p* = 0.0867). The CBPV prevalence was 19.10% in clinically affected honeybees, and no positive sample was detected in healthy honeybees (Table 4, Figure 2). The DWV prevalence was statistically significantly higher in affected honeybees (69.66%) than in healthy honeybees (39.81%) (*p* ˂ 0.0001); a strong statistical difference in viral loads was also found between healthy and affected honeybees with mean viral loads 6.37 log_10_ viral copies/bee for healthy and 8.69 log_10_ viral copies/bee for affected honeybees (*p* ˂ 0.0001). The LSV3 prevalence of positive samples was higher in healthy honeybees (67.59%) than in affected honeybees (48.31%) (*p* = 0.0087). Mean viral load in healthy honeybees (6.44 log_10_ viral copies/bee) was also higher than in affected honeybees (6.03 log_10_ viral copies/bee), but the difference was not statistically significant (*p* = 0.5484). For SBV, the detected prevalence was higher in healthy honeybees (35.19%) than in affected honeybees (22.47%), but the difference was only marginally statistically significant (*p* = 0.0601). However, the viral loads were statistically significantly higher in affected honeybees (6.92 log_10_ viral copies/bee) than the mean viral loads (4.45 log_10_ viral copies/bee) in healthy honeybees (*p* ˂0.0001) (Table 4, Figure 3).

### 3.3. The Average Viral Load Variations in Positive Samples of Healthy Honeybees Collected in a 10-Month Period

For ABPV, significant variations of viral loads were detected between four locations (Locations 1, 2, 3, 4). The lowest viral loads (from 4.68 to 5.81 log_10_ copies/bee) were identified in Locations 2 and 3, while the highest viral load (to 11.69 log_10_ copies/bee) of ABPV was detected in Location 4 in November and December 2018 and October 2019 (Figure 4). The detected viral loads for BQCV in individual apiary were stable over the 10-month sampling period but different and strongly co-related with the location (mean log_10_ copies/bee: Location 1: 6.50, Location 2: 6.78, Location 3: 6.24, Location 4: 7.92). The highest mean viral loads for BQCV were found in Location 4, while the maximal viral load was found in Location 3 in July 2019 (11.95 log_10_ copies/bee), and the lowest viral load was also detected in Location 3 in September 2019 (4.62 log_10_ copies/bee) (Figure 4). The monthly viral load variations for CBPV were not possible to present, while all samples in healthy honeybees were negative. The highest variations in DWV viral loads were observed in Location 4; increased viral loads were detected in October and November 2018 with a peak in December 2018 (13.17 log_10_ copies/bee), then DWV viral loads were decreasing (February, March, April 2019) and started increasing in May 2019 and then again in autumn months (Figure 4). The detected viral loads for LSV3 in four locations were between 3.97 log_10_ and 11.29 log_10_ copies/bee, while the highest viral loads were detected in Location 4 (Figure 5). Low variations in viral loads were observed for SBV in four locations. The detected viral loads for SBV varied between 3.95 log_10_ and 6.08 log_10_ copies/bee (Figure 5).

## 4. Discussion

The TaqMan RT-qPCR assays for quantification of LSV3 and SBV with newly designed primers, probes, and standards were constructed, while only a limited number of assays for quantification of these two viruses were described before. The results of this study confirmed that both new assays provide reliable results for detection and quantification of honeybee samples in the range from 2 log_10_ to 12 log_10_ copies/bee, similar to previously described assays of the European Union Reference Laboratory for Honeybee Health (EURL ANSES, Sophia Antipolis, France) [18,30,41].

The identified strains of LSV are genetically highly diverse, and there are more than eight different lineages of LSV known until now; three different lineages were identified in Slovenia [4,28]. The recent phylogenetic comparison of partial RNA-dependent RNA polymerase gene region (557 nucleotides) revealed 75.92% of honeybee samples and 17.07% of bumblebee samples as positive for LSV. Clustering of 26 Slovenian LSV strains revealed three genetic lineages: in the LSV lineage 1 (1 sample, 3.85%), the LSV lineage 2 (10 samples, 38.46%), and the LSV lineage 3 (15 samples, 57.69%) [28]. The high detected prevalence of LSV and limited data regarding this virus were the reasons for the development of a new fast, reliable RT-qPCR assay for the quantification of LSV3 from field samples as this was the strain with the highest prevalence from all the detected LSV lineages in Slovenia. Because of previously identified high diversity among different field strains of LSV, it was impossible to design one RT-qPCR assay for the quantification of all lineages of LSV. Nevertheless, important new data regarding the assay performance, the prevalence and the viral loads of LSV3 were obtained and successfully presented for the first time in healthy and clinically affected honeybee field samples.

The previously developed TaqMan assays for ABPV, BQCV, CBPV, and DWV showed good performance and reliability also for the detection and quantification of a broad genetic range of previously determined Slovenian honeybee viral strains [4,14,28]. The newly developed TaqMan RT-qPCR assay for detecting and quantifying SBV from this study is a good alternative and improvement of previously described assays, in terms of SYBR Green or MGB-probes [30,42,43,44], while SYBR Green also provides non-specific detection, and MGB-probes are not as stable as TaqMan probes. Because of this, TaqMan probes are the most frequently used in routine diagnostic laboratories for diagnostics of infectious diseases [45,46,47]. For further evaluation, the implemented assays should be tested on field samples from different countries to compare their usefulness for diverse strains and clusters of honeybee viruses. These implemented RT-qPCR assays are not suitable only for detecting and quantifying six honeybee viruses in adult honeybees but also in their brood, varroa mites, honeybee products, and other wild pollinators.

Significant differences were observed in viral prevalence for six honeybee viruses in clinically healthy honeybees in a 10-month period of sampling, strongly dependent on sampling location (i.e., apiary) (Table 4). This study also confirmed how important factors are the single apiary and their management in honeybee virus prevalence and viral load. The school apiary located in the city centre of Ljubljana (Location 4) was under regular veterinary monitoring but rather poorly managed; new colonies were added to the apiary while serving mainly for educational and experimental activities. Due to this, honeybee colonies were under constant stress and, as expected, the detected prevalence of six honeybee viruses in this apiary was the highest. Moreover, in seemingly healthy bees, the immunity could be affected, making the perfect conditions for viral replication supported with significantly higher viral loads of ABPV, BQCV, DWV, and LSV3 than in the other three locations. The apiary on Location 1 was the best managed, owned by a private company dealing with intensive honey production. Here, varroa mite was held under strict control as well as the viral infections with low observed prevalence. Thus, the results of this study confirmed some of the viral pathogens, such as ABPV, CBPV, DWV, and SBV, which were detected in selected honeybee colonies, but mainly with low viral loads. The apiary in Location 3 was placed in well-preserved nature with extensive honey production and isolated from other apiaries where not much contact with foreign bees was possible. This apiary was also well managed, using organic acids to control varroa and the results of virus prevalence and detected varroa loads support a favourable situation regarding viral infections during 10 months of sampling. The apiary in Location 2 was a small, hobby apiary, as most Slovenian beekeeper’s apiaries are, consisting of up to 10 honeybee colonies, but this apiary was well managed and with a favourable situation regarding varroa and viral infections. Although samples were collected from clinically healthy honeybee colonies, significant variations were observed in the apiaries with different management from four different locations when comparing the prevalence and viral loads of six honeybee viruses during 10 months of sampling. In general, the better the management and the more remote the location of the apiary, the less likely the colonies are to be affected by severe viral infections with clinical symptoms.

When comparing the prevalence and detected viral loads in clinically healthy and affected colonies, statistically significant differences were observed for ABPV, CBPV, DWV, and SBV. This observation is supported by a previous study from France, in which various thresholds for different honeybee viruses were suggested [30]. The results from clinically affected colonies could be more diverse, since they were collected from more diverse locations and apiaries, which must be taken into account when interpreting the results, but we consider also the monthly sampling and changes during time do present enough versatile results in healthy honeybees and this should not lead to misinterpretation.

The previously determined threshold for ABPV viral load as 5 log_10_ genome copies/bee [30] was confirmed as completely acceptable in our study, because the majority of viral loads of affected colonies were above this threshold, and only samples of healthy honeybees collected from Location 4 were above this value.

From the results of our study, the mean detected viral load of BQCV for both healthy and affected colonies was around 7 log_10_ genome copies/bee, which is slightly below the previously suggested threshold for BQCV (8 log_10_ genome copies/bee) [30].

The previously determined threshold for CBPV set on 6 log_10_ genome copies/bee [18] is also supported by our study’s results, on the basis of 89 samples of clinically affected and 108 healthy honeybees.

The previously determined threshold for DWV viral load (6 log_10_ genome copies/bee) [30] was also confirmed with observations in our study in healthy and clinically affected honeybees. Although in six (66%) of nine DWV positive samples collected from healthy honeybees on Location 4, the viral loads were above this threshold, this is rather more related to specifics of this apiary and high detected prevalence of different viruses (Table 4) and management, as already described above.

The average of detected viral loads for LSV3 in healthy and clinically affected honeybee colonies was almost the same and around 6 log_10_ genome copies/bee, which are the first viral load data for LSV3 from field samples. According to the observation of this study and the previously observed prevalence [4,28], a high frequency of LSV3 transmission and replication is present in field samples but with probably low virulence and limited impact on the health status of honeybee colonies. The determined viral load value for LSV3 in healthy and clinically affected honeybees was around 6 log_10_ genome copies/bee in both groups of samples; thus, the threshold value for LSV3 need not be set now.

The suggested threshold value for SBV at 9 log_10_ genome copies/bee [30] was set higher than the observation of this study. However, in the French study, the brood samples were presented, which are known to harbour enormous amounts of viral copies [29], and this threshold could be correct for the infected larvae but not for the adult honeybees [30]. The direct comparison of healthy and clinically affected colonies in this study showed clear differentiation between both groups because low SBV viral copy numbers were detected (mean 4.45 log_10_ copies/bee) in adult clinically healthy honeybees. In contrast, in clinically affected adult honeybees, the mean viral load of SBV-positive samples was 6.92 log_10_ viral copies/bee, suggesting that adult honeybees with detected viral loads around 7 log_10_ genome copies/bee are a good indicator of infection with SBV in the brood. This may be related to the detection of contaminated adult honeybees collected in affected honeybee colonies during the cleaning of SBV-infected cells.

The correlation between the determined viral loads and the clinical signs of viral infections (dead honeybees in front of hives, colony collapse, weak colonies, trembling, paralysis, loss of colour and hair, wing deformities, or shorter life span in colonies), recognised by our veterinary specialists, is supported with laboratory detection and quantification of four of six honeybee viruses. The constant honeybee health status monitored in Slovenia is led by the local veterinary specialists, organised within eight local units of the National Veterinary Institute, Veterinary Faculty, University of Ljubljana, all of which have received, in addition to basic university veterinary school, specific education about honeybee diseases and management. These veterinary specialists have direct and constant contacts with beekeepers, access to different kinds of field samples, and several years of field experience, including cooperation with different laboratories. In this study, the honeybee colonies were selected, and samples were collected from 89 different apiaries throughout Slovenia according to the decision of these veterinary specialists, meaning that the samples originated from clinically affected honeybee colonies. The first quantitative results for Carniolan honeybees and laboratory determination of thresholds and viral loads for ABPV, CBPV, DWV, and SBV strongly support clinical observations (justified suspicion of viral infection) of the veterinary experts when comparing and interpreting the results for healthy and clinically affected honeybee colonies. No major differences in viral loads were observed for BQCV and LSV3, supporting some of the previous observations that these two viruses are less pathogenic but frequently detected in clinically healthy and affected honeybee colonies [13,28].

The determination of viral load variations collected over 10 consecutive months in four different locations (apiaries) in healthy honeybees in Slovenia present important information of high variations in prevalence and viral load determination for positive samples. Although limited previously published studies exist, using conventional RT-PCR detection assays, without quantification [31], this study shows not only the prevalence variation for positive samples within four different types of apiaries but also variations of selected colonies’ viral loads for ABPV, BQCV, DWV, LSV3, and SBV during a one-year period, including winter months. Winter sampling of honeybees was rarely practised and was not possible in all selected apiaries in our study because of cold weather or disturbing the bees during their hibernation. Nevertheless, the results of our study provide important new data on possibly increasing viral loads for ABPV and DWV in selected colonies during the winter months, which were not presented before.

## 5. Conclusions

The performance of six different TaqMan RT-qPCR methods designed to quantify honeybee viruses (ABPV, BQCV, CBPV, DWV, LSV3, and SBV) in healthy and clinically affected honeybee colonies showed very reliable and accurate results. The TaqMan RT-qPCR methods for quantification of LSV3 and SBV were newly constructed and put on the same level as the quantification methods of the European Union Reference Laboratory for Honeybee Health in the aspect of reliability and offered them for future diagnostic and scientific use. The statistically significant correlations between viral loads for ABPV, BQCV, CBPV, and SBV, and clinical signs in adult honeybees were confirmed. Previously set threshold viral load levels between healthy and clinically affected honeybees for ABPV, BQCV, and DWV proved to be correct, and additional thresholds for CBPV, SBV, and LSV were suggested. The apiary location and management may have a much more important role on virus prevalence viral load in clinically healthy honeybee colonies than was previously suggested.

## Figures and Tables

**Figure 1 viruses-13-01340-f001:**
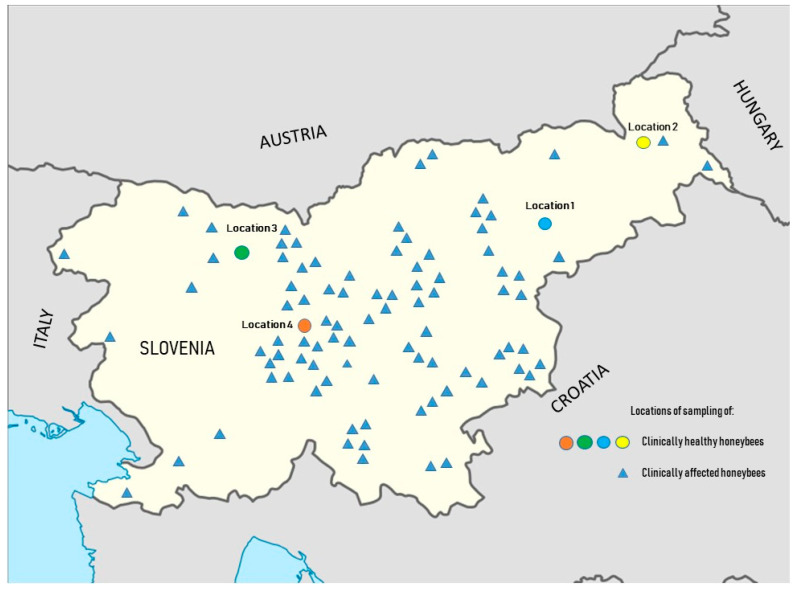
Locations of sample collection of healthy honeybees (Location 1—Štajerska, Location 2—Prekmurje, Location 3—Gorenjska, Location 4—Ljubljana) marked with in different colours and clinically affected honeybees marked with ▲ from throughout Slovenia.

**Figure 2 viruses-13-01340-f002:**
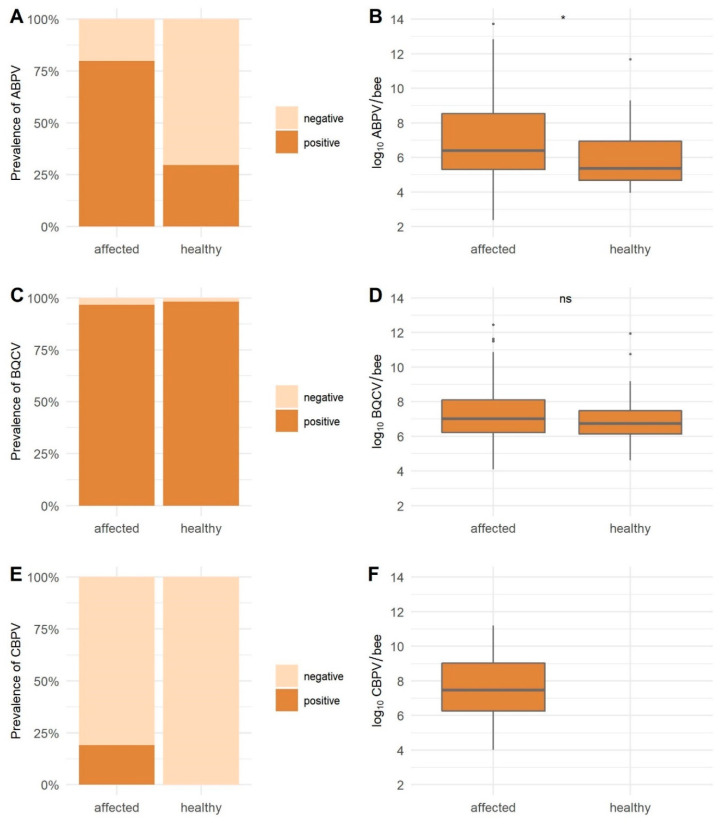
The detected prevalence (**A**,**C**,**E**) and the average viral load (**B**,**D**,**F**) for ABPV, BQCV, and CBPV in healthy and clinically affected honeybees. * = statistically significant difference; **** = strong statistically significant difference; ns = no statistically significant difference.

**Figure 3 viruses-13-01340-f003:**
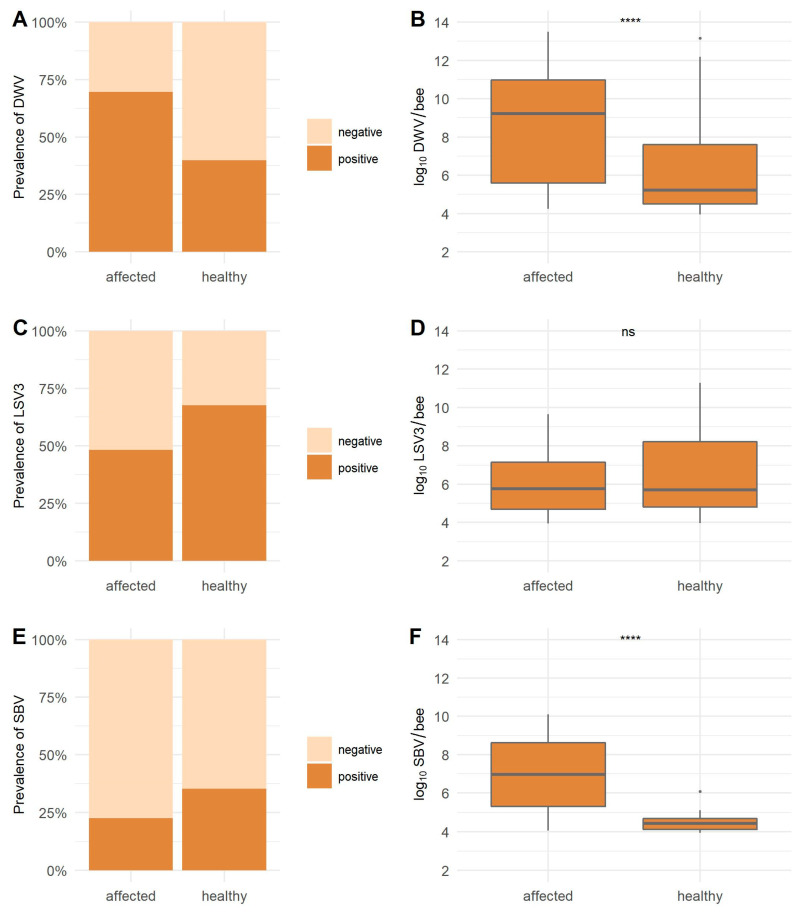
The detected prevalence (**A**,**C**,**E**) and the average viral load (**B**,**D**,**F**) for DWV, LSV3, and SBV in healthy and clinically affected honeybees. * = statistically significant difference; **** = strong statistically significant difference; ns = no statistically significant difference.

**Figure 4 viruses-13-01340-f004:**
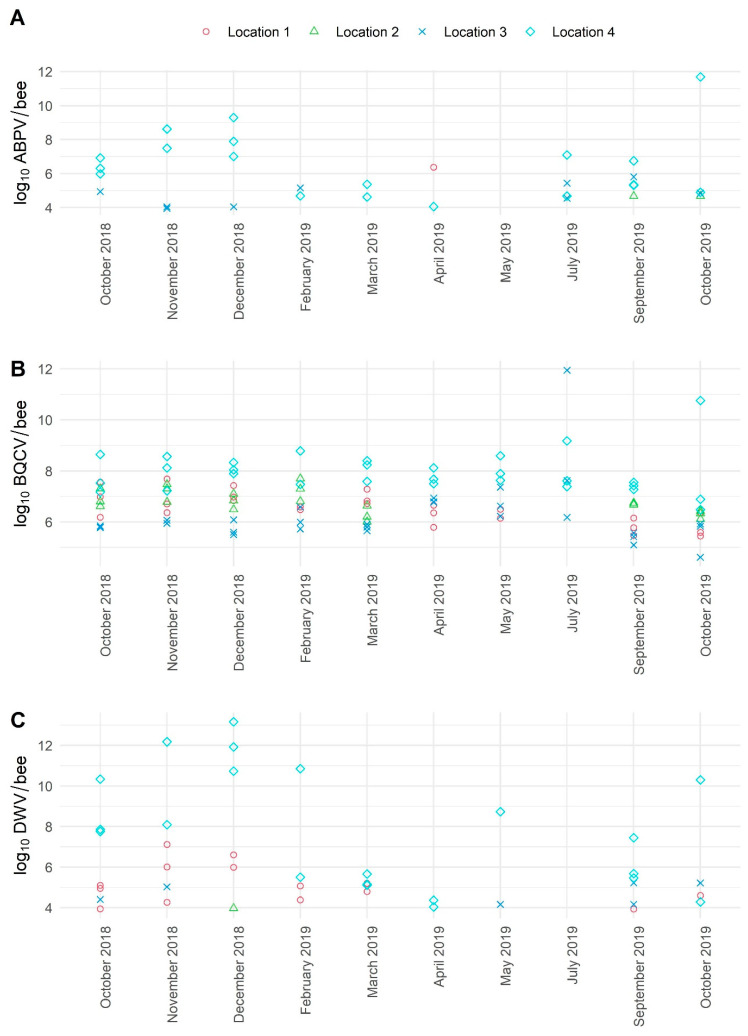
The detected average viral loads of positive samples for ABPV (**A**) (no. of samples: 109, no. of positive samples: 32), BQCV (**B**) (no. of samples: 109, no. of positive samples: 106), and DWV (**C**) (no. of samples: 109, no. of positive samples: 43) of healthy adult honeybees during the 10 months of sampling. One symbol represents one positive honeybee sample consisting of 10 honeybees.

**Figure 5 viruses-13-01340-f005:**
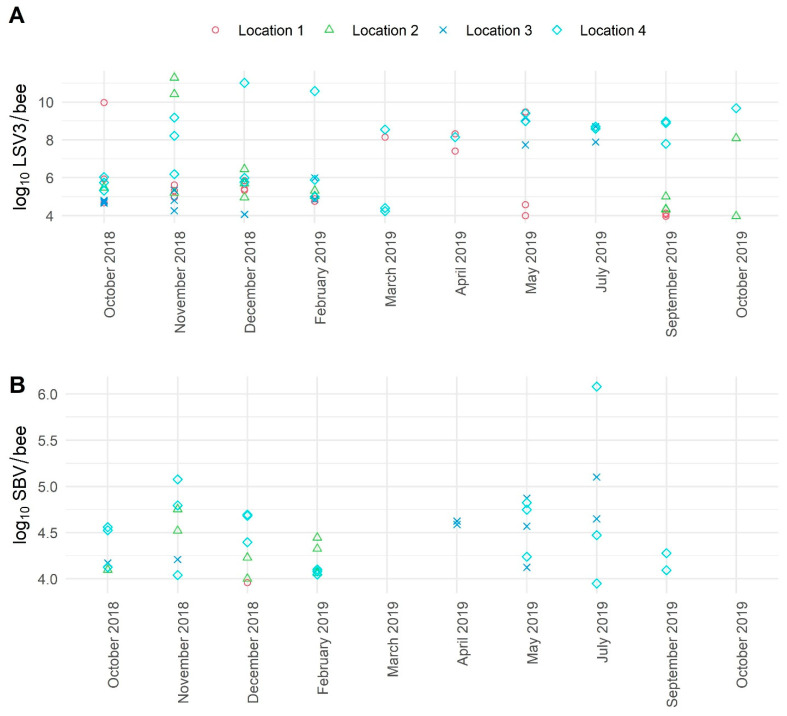
The detected average viral loads of positive samples for LSV3 (**A**) (no. of samples: 109, no. of positive samples: 73) and SBV (**B**) (no. of samples: 109, no. of positive samples: 38) of healthy adult honeybees during the 10 months of sampling. One symbol represents one positive honeybee sample consisting of 10 honeybees.

**Table 1 viruses-13-01340-t001:** Primers and TaqMan probes used for RT-qPCR assays.

Virus	Forward Primer (5′-3′)	Reverse Primer (5′-3′)	Probe (5′-3′)	Reference	Primer Concentration	Probe Concentration
ABPV	CATATTGGCGAGCCACTATG	CTACCAGGTTCAAAGAAAATTTC	(6-Fam) ATAGTTAAAACAGCTTTTCACACTGG (Tamra)	[32]	800 mM	100 nM
BQCV	GGTGCGGGAGATGATATATGGA	GCCGTCTGAGATGCATGAATAC	(6-Fam) TTTCCATCTTTATCGGTACGCCGCC (Tamra)	[34]	320 nM	200 nM
CBPV	CGCAAGTACGCCTTGATAAAGAAC	ACTACTAGAAACTCGTCGCTTCG	(6-Fam) TCAAGAACGAGACCACCGCCAAGTTC (Tamra)	[12]	320 nM	200 nM
DWV	GCGGCTAAGATTGTAAATGTC	GTGACTAGCATAACCATGATTA	(6-Fam) CCTTGACCAGTAGACAGCATC (Tamra)	[30]	350 nM	100 nM
LSV3	GTACCTACACTCTTCCATGCAC	CCAGCTAAGGAGGCGTAAC	(Cy5) TCACCCACATCATTGAGCCAGGT (BHQ2)	This study	200 nM	100 nM
SBV	AGCCAGTGATAGATGCTC	AAATACTCCCGCCAAATCAC	(6-Fam) TGGCTCATCTGGGATCACAATTTCC (Tamra)	This study	200 nM	50 nM

**Table 2 viruses-13-01340-t002:** Standards used for quantification of six honeybee viruses by RT-qPCR assays.

Standard Name	Plasmid Vector (Supplier)	Cloned Sequence: Nucleotide Position	Sequence Length	Virus (GenBank Accession Number)	Reference
pB2	pCR II Topo (Invitrogen, Waltham, MA, USA)	8115 to 8512	397	ABPV (AF126050)	[35]
pNC1-4	pGEM-T Easy (Promega, Madison, WI, USA)	7850 to 8550	700	BQCV (AF183905)	[36]
pAb2	pGEM-T Easy (Promega, Madison, WI, USA)	2260 to 3059	799	CBPV (EU122229)	[37]
pC1	pCR II Topo (Invitrogen, Waltham, MA, USA)	4240 to 4659	419	DWV (AY292384)	[37]
LSV3	LSV3-SI (Sigma-Aldrich, Steinheim, Germany)	2021 to 2265	244	LSV3 (KY465717)	This study
SBV	SBV-SI (Sigma-Aldrich, Steinheim, Germany)	5030 to 5368	338	SBV (MG545287)	This study

**Table 3 viruses-13-01340-t003:** The performance of RT-qPCR assays for quantification of ABPV, BQCV, CBPV, DWV, LSV3, and SBV by using 10-fold standard dilutions and three repetitions of each assay.

ABPV	Repeat 1	Repeat 2	Repeat 3	Mean	SD
Intercept	43.96	48.47	45.35	45.93	±2.54
Slope	−3.084	−3.345	−3.150	−3.193	±0.152
Efficiency	111.0	99.0	107.7	105.9	±6.9
R^2^	0.995	0.978	0.982	0.985	±0.010
BQCV	Repeat 1	Repeat 2	Repeat 3	Mean	SD
Intercept	42.77	41.80	41.77	42.11	±0.66
Slope	−3.362	−3.102	−3.036	−3.167	±0.195
Efficiency	99.2	110.1	113.5	107.6	±8.4
R^2^	0.993	0.994	1.000	0.996	±0.004
CBPV	Repeat 1	Repeat 2	Repeat 3	Mean	SD
Intercept	48.35	48.04	46.05	47.48	±1.43
Slope	−3.579	−3.432	−3.358	−3.456	±0.123
Efficiency	90.3	95.9	98.5	94.9	±4.6
R^2^	0.995	0.996	1.000	0.997	±0.003
DWV	Repeat 1	Repeat 2	Repeat 3	Mean	SD
Intercept	42.74	42.65	40.38	41.92	±1.54
Slope	−3.334	−3.059	−3.011	−3.135	±0.199
Efficiency	99.5	112.3	114.8	108.9	±9.37
R^2^	0.981	0.998	0.997	0.992	±0.011
LSV 3	Repeat 1	Repeat 2	Repeat 3	Mean	SD
Intercept	42.52	41.04	39.58	41.05	±1.47
Slope	−3.395	−3.607	−3.460	−3.487	±0.120
Efficiency	97.0	89.9	94.5	93.8	±3.9
R^2^	0.997	0.998	0.995	0.997	±0.002
SBV	Repeat 1	Repeat 2	Repeat 3	Mean	SD
Intercept	41.08	40.29	37.68	39.68	±2.00
Slope	−3.616	−3.517	−3.264	−3.466	±0.202
Efficiency	89.0	92.5	102.5	94.7	±7.8
R^2^	0.997	0.999	1.000	0.999	±0.002

**Table 4 viruses-13-01340-t004:** The obtained prevalence for ABPV, BQCV, CBPV, DWV, LSV3, and SBV in healthy and clinically affected honeybee colonies detected by RT-qPCR assays.

Location No. No. of Samples	ABPV	BQCV	CBPV	DWV	LSV3	SBV
Location 1—healthy *n* = 27	3.70	96.30	0.00	55.56	74.07	3.70
Location 2—healthy *n* = 21	9.52	100	0.00	4.76	71.43	38.10
Location 3—healthy *n* = 30	30.00	100	0.00	20.00	46.67	30.00
Location 4—healthy *n* = 30	66.67	96.67	0.00	70.00	80.00	66.67
Total—healthy *n* = 108	29.63	98.15	0.00	39.81	67.59	35.19
Total—affected *n* = 89	79.78	96.63	19.10	69.66	48.31	22.47

## Data Availability

Not applicable.

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
