# Peer review of "The Comparison of Honeybee Viral Loads for Six Honeybee Viruses (ABPV, BQCV, CBPV, DWV, LSV3 and SBV) in Healthy and Clinically Affected Honeybees with TaqMan Quantitative Real-Time RT-PCR Assays"

_viruses, 2021, doi:10.3390/v13071340_

Round 1

Reviewer 1 Report

This manuscript represents a considerable amount of work and a great dataset. It will be of interest to readers working on honey bee viruses, and for readers with an interest on methodology on virus quantification. I think it is likely to be publishable but would first benefit from some degree of revision. My specific comments are as follows:

Lines 15-29. The abstract describes an interesting study. I’d like to see “Clinically affected” described better here. And I’d suggest a primary focus on the hypothesis testing between diseased and healthy colonies, rather than the focus on the methodology that currently occurs in the introduction.

Lines 124-151. The introduction is very long and could be cut. Some of the information is interesting and useful, such as with the description of the viruses. Other sections are less useful. The paragraph from lines 124-130 just says that a previous study a]has occurred in Slovenia and viruses were found. The following paragraph from lines 131-151 could be cut or placed in the methods.

Lines 181-188. The study contains a fantastic sample size. However, the ‘clinically affected’ hives were clearly suffering from a range of different viral or pathogen infections. Some had deformed wings indicating Deformed wing virus infections, while others were trembling indicating paralysis viruses. Consequently, grouping all ‘clinically infected’ samples together for comparison against healthy colonies doesn’t make a lot of sense. Could the authors try different groupings and analyses? I’d have preferred the healthy samples to have come from a wider variety of spatial localities and apiaries, as currently there will be more variation in the ‘clinically affected’ hives than in the healthy ones; perhaps this could be acknowledged more in the discussion.

Lines 200-210. The authors state that “Primers, TaqMan probes, and quantification standards were designed from multiple alignments of previously se-quenced LSV3 and SBV strains from Slovenia [4,28] and strains available in GenBank”. We’d probably like to use their assay too, but how specific do they think it would be to viral strains from Slovenia compared to others around the world?

Lines 236-239. I wasn’t quite sure how they used the standards here. More information could be presented and provided in the table captions. I don’t think figure 2 is needed.

Lines 267-270. The authors state that “Since the re-peated measures in the same healthy honeybee colonies were measurements of the same variables and we were not interested in the differences over time, we did not perform repeated measure analysis.” I think a lot of readers would be interested in a repeated analysis and the results it supplies (this sort of statement smells like a comment from a previous review).

Line 292. Use periods rather than commas in your R2 values for consistency. And on line 332.

Results. I think more statistical analysis and interpretation is needed here. For example, the authors state that “The LSV3 prevalence of positive samples was higher in healthy honeybees (67.59%) than in affected honeybees (48.31%); mean viral load in healthy honeybees (6.44 log10 viral copies/bee) was also higher than in affected hon-eybees (6.03 log10 viral copies/bee)”. The graphs don’t really support that, the levels of infection look similar. I think the authors need to be able to say if they were statistically different or not. Virus abundance is often not normally distributed and I recommend permutation tests.

Figures 5 &6. These graphs really do make me think that the authors are interested in samples over time, and thus a repeated measures analysis might be appropriate.

Lines 419-447. This paragraph talks about individual apiaries and their management. I think broader trends in the data will be more interesting to most readers. I’d recommend smaller succinct paragraphs (I.e. not lines 448-487). Perhaps more focus in the discussion on an international context for the results would make the article more appealing too.

References. A relevant set of references is included. Please ensure you italicise genus and species names, and standardise the capitalisation of publication titles.

Reviewer 2 Report

I was impressed by this paper as a wide ranging analysis of bee virus prevalence in hives at a number of locations.  Molecular tools are being applied increasingly to the study of pathogens, none more so than in the current Covid-19 pandemic.  It is good to see qPCR being applied to the study of virus loads in bee population with a view to determining how the levels of these pathogens vary according to site and also hive management.  I thought one of the most interesting aspects of the work was the correlation between virus load and poor hive maintenance, which might suggest ways of mitigating the problem if colony collapse. It was also interesting to see how in some instances (e.g. BQCV) that virus load could be high even in healthy insects.  This mirrors what has been seen in lepidopteran populations of insects where baculovirus/cypovirus occurrence does not always mean the death of the host.

Clearly, we still have a lot to learn about bee decline but this paper will help others devise strategies to monitor populations and keep them healthy.

Author Response

We are really honored and proud to receive such a nice review. This gives us so much motivation for further research, so thank you very much.